# Decision Tree Integration Using Dynamic Regions of Competence

**DOI:** 10.3390/e22101129

**Published:** 2020-10-05

**Authors:** Jędrzej Biedrzycki, Robert Burduk

**Affiliations:** Department of Systems and Computer Networks, Wroclaw University of Science and Technology, 50-370 Wroclaw, Poland; jedrzej.biedrzycki@pwr.edu.pl

**Keywords:** decision tree, random forest, majority voting, classifier ensemble, classifier integration

## Abstract

A vital aspect of the Multiple Classifier Systems construction process is the base model integration. For example, the Random Forest approach used the majority voting rule to fuse the base classifiers obtained by bagging the training dataset. In this paper we propose the algorithm that uses partitioning the feature space whose split is determined by the decision rules of each decision tree node which is the base classification model. After dividing the feature space, the centroid of each new subspace is determined. This centroids are used in order to determine the weights needed in the integration phase based on the weighted majority voting rule. The proposal was compared with other Multiple Classifier Systems approaches. The experiments regarding multiple open-source benchmarking datasets demonstrate the effectiveness of our method. To discuss the results of our experiments, we use micro and macro-average classification performance measures.

## 1. Introduction

Multiple Classifier Systems (MCS) are a popular approach to improve the possibilities of base classification models by building more stable and accurate classifiers [1]. MCS are one of the major development directions in machine learning [2,3]. MCS proved to have a significant impact on the system performance, therefore they are used in many practical aspects [4,5,6,7].

MCS are essentially composed of three stages: generation, selection and fussion or integration. The aim of the generation phase is to create basic classification models, which are assumed to be diverse. This goal is achieved, inter alia, by methods of dividing the feature space [8]. In the selection phase, one classifier (the classifier selection) or a certain subset of classifiers is selected (the ensemble pruning) learned at an earlier stage. The fusion or the integration process combines outputs of base classifiers to obtain an integrated model of classification, which is the final model of MCS. One of the commonly used methods to integrate base classifiers’ outputs is the majority vote rule. In this method each base model has the same impact on the final decision of MCS. To improve the efficiency of MCS the weights are defined and used in the integration process. The use of weights allows to determine the influence of a particular base classifier on the final decision of MCS. The most commonly used approach to determining the weights uses probability error estimators or other factors [9,10,11]. A distance-weighted approach to calculating the weights is also often used in many problems, were the weights are determined [12,13,14]. In general, this approach is based on the query where the appropriate object is located. In this article, we use the feature subspace centroid in the definition of the distance-weighted approach.

There are, in general, two approaches to partition a dataset [15]. In horizontal partitioning the set of data instances is divided into a subset of datasets that are used to learn the base classifiers. Bagging bootstrap sampling to generate a training subset is one of the most used method in this type of datasets partitioning. In the vertical partitioning the feature set is divided into feature subsets that are used to learn the base classifiers. Based on vertical partitioning feature space the forest of decision trees was proposed in [16]. Contrary to the types of dataset partitioning mentioned above, the clustering and selection algorithm [17] is based on the clustering. After clustering, one classifier is selected for each feature subspace. In this algorithm the feature space partition is an independent process from the classifier selection process and precedes this selection. The non-sequential approach to clustering and selection algorithm was probesed in [18,19].

In this work, we propose a novel approach to determining the division feature space into the disjoint feature subspace. Contrary to the clustering and the selection method described above in our proposal the proces of partitioning the feature space follows base classifier learning. Additionally, the proposed approach does not use clustering to define a feature subspace. The partiotion of the feature space is defined by base classifier models, and exactly through their decision boundaries. According to our best knowledge the use of the decision boundary of base models for partitioning feature space is not represented in MCS. Finally, the centroids of proposed feature subspace are used in the weighted majority voting rule to define the final MCS decision.

Given the above, the main objectives of this work can be summarized as follows:
A proposal of a new partitioning of the feature space whose split is determined by the decision bonduaries of each decision tree node which is a base classification model.The proposal of a new weighted majority voting rule algorithm dedicated to the fusion of decision tree models.An experimental setup to compare the proposed method with other MCS approaches using different performance measures.


The outline of the paper is as follows: In Section 2 related works are presented. Section 3 presents the proposed approach to MCS fusion process. In Section 4 the experiments that were carried out and the discusion of the obtained results are presented. Finally, we conclude the paper in Section 6.

## 2. Related Works

Classifier integration using the geometrical representation has already been mentioned in [20]. Based on transformations in the geometrical space spread on real-valued, non-categorical features this procedure has proven itself to be more effective in comparison to others, commonly used integration techniques such as majority voting [21]. The authors have studied and proved the effectiveness of an integration algorithm based on averaging and taking median of values of the decision boundary in the SVM classifiers [22]. Next, two algorithms for decision trees were proposed and evaluated [23,24]. They have proven themselves to provide better classification quality and ease of use than referential methods.

Polianskii and Pokorny have examined a geometric approach to the classification using Voronoi cells [25]. Voronoi cells fulfill the role of the atomic elements being classified. Labels of the nearest training objects are assigned to the boundaries. The algorithms walks along the boundaries and integrates them with respect to the associated class. SVM, NN and random forest classifiers were used in evaluation.

The nearest neighbor algorithm can be used to test which Voronoi cell an object belongs to [26]. By avoiding the calculation of the Voronoi cells geometry, the test appears to be very efficient. On that basis a search lookup was described by Kushilevitz et al. [27]. A space-efficient data structure is utilized to find an approximately nearest neighbor in nearly-quadratic time with respect to the dimensionality.

However, the nearest neighbor algorithms are difficult. The number of prototypes needs to be specified beforehand. Using too many causes high computational complexity. Too few, on the other hand, can result in an oversimplified classification model. This matters especially when datasets are not linearly separable, have island–shaped decision space, etc. There are several ways to solve this problem that can be found in the literature. Applying Generalized Condensed Nearest Neighbor rule to obtain a set of prototypes is one of the possible solutions [28]. In this method a constraint is added, that each of the prototypes has to come from the training dataset. A different approach was proposed by Gou et al. [29]. Firstly, kNN algorithm is used to obtain a certain number of prototypes for each class. Afterwards the prototypes obtained in the first step are transformed by the local mean vectors. This results in a better representation of the distribution of the decision space.

Decision trees are broadly used due to their simplicity, intuitive approach and at the same time good efficiency. The way they classify the objects is by recursive partitioning of the classification space [30]. Although they have first appeared more than three decades ago [31], the decision tree algorithm and its derivatives are in use in a range of industry branches [32].

At some point it has been noticed that the local quality of each of the base classifiers is different. The classifier selection process was introduced in order to choose a subset of base classifiers that have the best classification quality over the region. The selection is called static, when the division can be determined prior to the classification. In the opposite scenario the new pattern is used to test the models’ quality [33]. Kim and Ko [34] have shown a greater improvement in the classification when using local confidence over averaging over the entire decision space.

Another approach to the classifier integration is by using a combination of weighting and local confidence estimation [35]. The authors noticed, that using only a subset of points limited to a certain area in the training process results in a better classification performance.

An article [36] discusses a variation of the majority voting technique. A probability estimate is computed as the ratio of properly classified validation objects over certain geometric constraints known a priori. Regions that are functionally independent from each other are treated separately. The proposed approach provides a significant improvement in the classification quality. The downside of this method is that the knowledge of the domain is necessary to provide a proper division. Additionally, the split of the classification space has to be done manually. The performance of the algorithm was evaluated using a retinal image and classification in its anatomic regions.

An improvement in the weighted majority voting classification can be observed for class–wise approach covered in [37]. For each label weights are determined separately for the objects in the validation dataset.

Random forest, introduced by Breiman in 2001 [38] is one of the most popular ensemble methods. It has proven itself to be very effective and many related algorithms were developed since then. Fernandez et al. studied 179 different classifications algorithms using 121 datasets [39]. The random forest outperforms most of the examined classifiers. It uses decision trees trained on distinct subsets of the training dataset. A majority voting over classifications of every model for an object under test is calculated as the final result.

Numerous algorithms involving gradient boosting and decision trees have emerged. Extreme Gradient Boosting (XGB) is an implementation of one of the most widely spread stacking techniques. It is used especially in machine learning competitions [40,41,42]. In theory subsequent decision trees are trained. Consecutive models minimize the value of loss function left by their predecessors [43]. Another implementation of Gradient Boosting Decision Tree designed with performance in mind, especially when working with datasets with many dimensions, is LightGBM [44]. Compared to the previous library, statistically no loss of performance in the classification is observed, but the process of training can be up to 20 times faster.

Vertical or horizontal partitioning can be used to force the diversity between base classifiers [30]. The datasets of extreme sizes are classified better using horizontal partitioning compared to bagging, boosting or other ensemble techniques [45].

## 3. Proposed Method

The proposed method is based on previous works of authors, but suggests a slightly different approach [23,24]. While the cited articles used static division into regions of competence, this paper presents an algorithm with a dynamic approach. The main goal of introducing the dynamically generated Voronoi cells is to achieve better performance than with other referential methods of the decision tree commitee ensembling: majority voting and random forest.

Before proceeding with the algorithm, datasets are normalized to the unit cube (every feature takes values in range of [0,1]) and two most informative features are extracted. Feature extraction is conducted using ANOVA method.

The first step of the presented algorithm is training a pool of base decision trees. To make sure the classifiers are different from one another, they are trained on the random subsets of the dataset. Having a commitee of decision trees trained, we are extracting rectangular regions that fulfill the following properties:
Their area is maximal.Every point they span is labeled with the same label by every single classifier (labels can differ across different classifiers). In other words regions span over the area of objects equally labelled by the classifier points.


In practice this means, that the entire space is divided along every dimension at all the split points of every decision tree. This way the regions are of the same class as indicated by every model.

Having the space divided into subspaces, midpoints are calculated. Let us denote by *S* the set of obtained subspaces and by $\left(x_{s,1,min};x_{s,1,max}\right)$ and $\left(x_{s,2,min},x_{s,2,max}\right)$ the range of subspace *s* along axis $x_{1}$ and $x_{2}$ respectively. The midpoint of subspace *s* will be denoted as $x_{s,mid}$. For every subspace and every label the weight is calculated using the following formula:
(1)$$f\left(\Psi_{i},s_{0}\right)=\frac{1}{\sigma }\sum_{s\in S}c_{s,\Psi_{i}}\left(1-d\left(x_{s_{0},mid},x_{s,mid}\right)\right)\delta \left(s_{0},s\right)+\frac{c_{s_{0},\Psi_{i}}}{2n}$$
where $d(p_{1},p_{2})$ is the euclidean distance between the points $p_{1}$ and $p_{2}$, $c_{s,\Psi_{i}}$ is the number of classifiers that classify the subspace *s* with the label $\Psi_{i}$, σ is the correction which purpose is to make the sum of weights equal 1 and $\delta (s_{0},s)$ is a function that returns 1 if $s_{0}$ and *s* are neighbors and 0 otherwise, i.e.,
(2)$$\delta \left(s_{0},s\right)=\left\{\begin{array}{cc} 1 & \mathrm{if}\;x_{s_{0},1,min}=x_{s,2,max}\;\mathrm{or}\;x_{s_{0},1,max}=x_{s,1,min}\\ \; & \mathrm{or}\;x_{s_{0},2,min}=x_{s,2,max}\;\mathrm{or}\;x_{s_{0},2,max}=x_{s,2,min}\\ 0 & \mathrm{otherwise} \end{array}\right.$$
It’s important to notice, that according to the Formula (Equation 2), δ(s,s)=0 for every subspace *s*. This is because the contribution of the subspace itself is reflected by the second summand of the Equation (Equation 1). The term 2n was chosen in the denominator, because then the subspace $s_{0}$ makes up half of the weight’s value, i.e.,
$$\sum_{i=1}^{n}\frac{c_{s_{0},\Psi_{i}}}{2n}=\frac{1}{2}$$


The process of obtaining subspaces is depicted in the Figure 1. Let us suppose, that all the base decision trees (colorful lines on subfigure a) are oriented in the same way - all the points below the decision boundary are classified by the given decision tree with a single label, different from all the objects above the line. As it was stated before, the competence regions are obtained by splitting the entire space at the splitpoints of all the decision trees (subfigure b). When calculating the weight of the label for each region, the region itself (filled with dark grey in subfigure c) together with its neighbors (lightgrey in subfigure c) are considered. Whereas the region itself contributes to half of its weight, contributions from every neighbor depend on the distance between its midpoint and the midpoint of the considered region. The entire procedure is presented in Algorithm 1.

**Algorithm 1:** Classification algorithm using dynamic regions of competence obtained from decision trees.
 **Input**: *K* – number of base classifiers ($\Psi_{1},\Psi_{2},\cdots ,\Psi_{K}$) **Output**: Integrated decision tree $\Psi_{i}$
**1** Normalize the dataset and select two most informative features.**2** Split dataset into K+1 subsets (*K* for training every base decision tree and 1 for testing).**3** Train base classifiers $\Psi_{1},\Psi_{2},\ldots ,\Psi_{K}$ and obtain their geometrical representation (splits and labels).**4** Divide the feature space using splits of all the decision trees.**5** For every region and every label calculate the weight using formula (Equation 1).**6** Classify every region by picking the label with the highest weight value.


## 4. Experimental Setup

The algorithm was implemented in Scala. Decision tree and random forest implementation from Spark MlLib were used [46]. The statistical analysis was performed with Python and libraries Numpy, Scipy and Pandas [47,48,49]. Matplotlib was used for plotting [50]. In Spark’s implementation the bottommost elements (leaves) are classified with a single label. The algorithm performs a greedy, recursive partitioning in order to maximize the information gain in every tree node. Gini impurity is used as the homogeneity measure. Continuous feature discretization is conducted using 32 bins. The source code used to conduct experiments is available online (https://github.com/TAndronicus/dynamic-dtree).

The experiments were conducted using open-source benchmarking datasets from repositories UCI and KEEL [51,52]. Table 1 describes the datasets used with the number of features, instances and imbalance ratio.

The imbalance ratio was given to stress the fact that accuracy is not a reliable metric when comparing the performance of the presented algorithm and the reference. It is calculated as the quotient of the count of objects with the major label (most frequent) and the objects with minor label (least common): $\mathrm{Imb}=\frac{\#\mathrm{major}\;\mathrm{class}\;\mathrm{objects}}{\#\mathrm{minor}\;\mathrm{class}\;\mathrm{objects}}$ [53]. If the value of Imb equals 1, then the dataset is balanced—all classes have the same amount of instances. The larger the value, the more imbalanced the dataset is. Some of the datasets are highly imbalanced, because of the low imbalance ratio, so other metrics other than average accuracy should be considered when comparing the performance of classifiers. The reason is explained in the following example. Suppose Imb=9 for a binary classification problem. When a classifier labels all the test objects with the label of the major class, its accuracy is $\mathrm{ACC}=\frac{9}{9+1}=90\%$. In the parentheses, together with the names, abbreviations of the datasets names were placed by which they will be further referenced for brevity.

The experiments were conducted according to the procedure described in Section 3 and repeated 10 times for each hyperparameter set. Together with integrated classifiers, referential methods were evaluated: majority voting of the base classifiers and random forest. The results were averaged. K=3 was taken as the number of base classifiers.

## 5. Results

The purpose of the experiments was to compare the classification performance measures obtained by the proposed algorithm (with the subscript *i*) with the known methods as references: majority voting (subscript mv) and random forest (subscript rf). The experiments were conducted 10 times for each setup and the results were averaged. Because we conducted experiments on the multiclass datasets, as the classification evaluation metrics we use micro- and macro-average precision, recall and F-score which is the harmonic mean of precision and recall. For this reason F-score takes both false positives and false negatives into account. Additionally, we present the results for overall accuracy. The F-score was computed alongside the accuracy because of the high imbalance of multiple datasets used, as it was indicated in Section 4. The F-score describes the quality of a classifier much better than the overall accuracy for the datasets with a high imbalance ratio and gives a better performance measure of the incorrectly classified cases than the overall accuracy. Accuracy can be in this case artificially high. The metrics are calculated as defined in [54]. In the Table 2 the results are gathered: average accuracy, micro- and macro-average F-score, while the Table 3 and Table 4 show results for micro- and macro-average respectively. Together with the mentioned metrics, Friedman ranks are presented in the last row – the smaller the rank, the better the classifier performs. However, it should be noted that for micro-average performance measures the result obtained for precision and recall are the same. This result is justified by the micro-averaging disadvantage, because for the frequent single-label per instance problems ${\mathrm{Precision}}_{\mu }={\mathrm{Recall}}_{\mu }$ [55].

## 6. Discussion

For the proposed weighting method of the decision tree integration all the calculated classification performance measures are better than of the referrential algorithms as indicated by the Friedman ranks. This statement holds true for all the classification performance measures that have been used. Post-hoc Nemenyi test after Friedman ranking requires the difference in ranks of 0.38 to define a significant statistical difference between the algorithms. For ${\text{F-score}}_{\mu }$ performance measure this condition is met, which means that the proposed method $\Psi_{i}$ achieves statistically better results than the reference methods $\Psi_{rf}$ and $\Psi_{mv}$. Whereas for ${\text{F-score}}_{M}$ performacne measure there is no such property as shown in the Figure 2. The micro-average measure counts the fraction of instances predicted correctly across all classes. For this reason the micro-average can be a more useful metric than macro-average in the class imbalance dataset. Thus, the results show that the proposed method improves the classification results, in particular of imbalanced datasets. This conclusion is confirmed by the values obtained for other performance measures. And so for micro-avarage precision and recall the difference in ranks betwenn $\Psi_{i}$ and $\Psi_{rf}$ indicated the statistical differences in the results. The corresponding difference for $\Psi_{i}$ and $\Psi_{rf}$ algorithms is very close to being able to state the statistical differences in the results because it is equal 0.36 (see Table 3). In case of macro-avarage precision the obtained results do not indicate significant difference in this performance measure. Whereas for macro-avarage recall (see Table 4) the obtained avarage Friedman ranks are equal for $\Psi_{i}$ and $\Psi_{rf}$ algorithms.

## 7. Conclusions

This paper presents a new approach for determining MCS. Contrary to the clustering and selection method we propose that the feature space partition is based on decision bonduaries defined by base classifier models. It means that we propose to use learned base classification models instead of clustering to determine the feature subspace. The centroids of the proposed feature subspace are used in the weighted majority voting rule to define the final MCS decision. In particular, a class label prediction for each feature subspace is based on adjacent feature subspaces.

The experimental results show that the proposed method may create an ensemble classifier that outperforms the commonly used methods of combining decision tree models—the majority voting and RF. Especially, the results show that the proposed method statistically improves the classification results measured by the mico-avarage $F_{1}$ classification performance measure.

According to our best knowledge the use of the decision boundary of base models to partition feature space is not represented in MCS. On the other hand, the proposed approach has also a drawback, because it uses geometrical centroids of defined feature subspaces. Consequently, our future research needs to be aimed at finding centroids of objects belonging to particular feature subspaces. Additionally, we can consider another neighborhood of a given feature subspace necessary to determine the decision rule. This neighborhood may, for example, depend on the number of objects in particular the feature subspace.

## Figures and Tables

**Figure 1 entropy-22-01129-f001:**
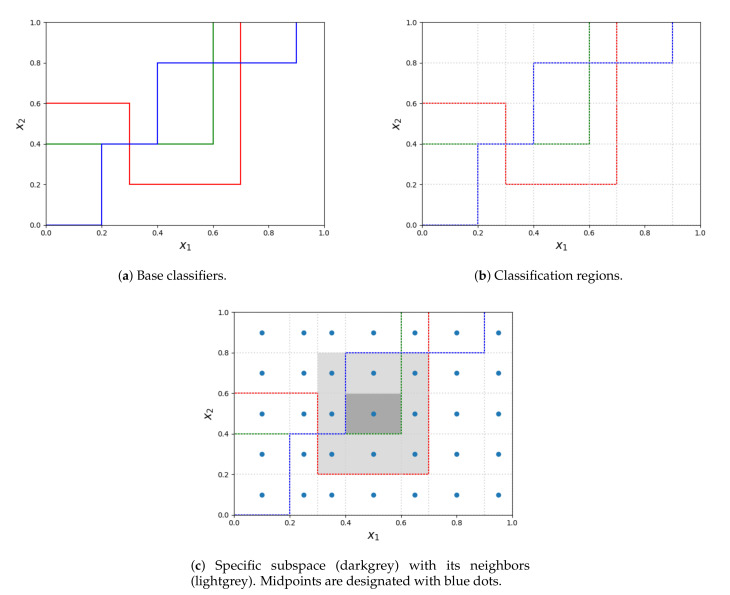
The process of extracting subspaces from base classifiers and determining neighbors for a subspace.

**Figure 2 entropy-22-01129-f002:**
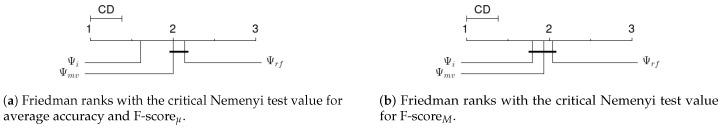
Friedman ranks for calculated metrics together with Nemenyi critical values.

**Table 1 entropy-22-01129-t001:** Descriptions of datasets used in experiments (name with abbreviation, number of instances, number of features, imbalance ratio).

Dataset	#inst	#f	Imb
Indoor Channel Measurements (aa)	7840	5	208.0
Appendicitis (ap)	106	7	4.0
Banana (ba)	5300	2	5.9
QSAR biodegradation (bi)	1055	41	2.0
Liver Disorders (BUPA) (bu)	345	6	1.4
Cryotherapy (c)	90	7	1.1
Banknote authentication (d)	1372	5	1.2
Ecoli (e)	336	7	71.5
Haberman’s Survival (h)	306	3	2.8
Ionosphere (io)	351	34	1.8
Iris plants (ir)	150	4	1.0
Magic (ma)	19,020	10	1.0
Ultrasonic flowmeter diagnostics (me)	540	173	1.4
Phoneme (ph)	5404	5	2.4
Pima (pi)	768	8	1.9
Climate model simulation crashes (po)	540	18	10.7
Ring (r)	7400	20	1.0
Spambase (sb)	4597	57	1.5
Seismic-bumps (se)	2584	19	14.2
Texture (te)	5500	40	1.0
Thyroid (th)	7200	21	1.0
Titanic (ti)	2201	3	2.1
Twonorm (tw)	7400	20	1.0
Breast Cancer (Diagnostic) (wd)	569	30	1.7
Breast Cancer (Original) (wi)	699	9	1.9
Wine quality – red (wr)	1599	11	68.1
Wine quality – white (ww)	4898	11	439.6
Yeast (y)	1484	8	92.6

**Table 2 entropy-22-01129-t002:** Average accuracy and f-scores for the random forest, the majority voting and the proposed algorithm together with Friedman ranks.

	Average Accuracy			F-Score_*μ*_			F-Score_*M*_		
Dataset	$\Psi_{\mathrm{mv}}$	$\Psi_{\mathrm{rf}}$	$\Psi_{i}$	$\Psi_{\mathrm{mv}}$	$\Psi_{\mathrm{rf}}$	$\Psi_{i}$	$\Psi_{\mathrm{mv}}$	$\Psi_{\mathrm{rf}}$	$\Psi_{i}$
aa	0.917	0.918	0.919	0.469	0.474	0.477	0.196	0.192	0.176
ap	0.853	0.812	0.863	0.853	0.812	0.863	0.676	0.559	0.692
ba	0.789	0.808	0.815	0.683	0.712	0.722	0.483	0.493	0.502
bi	0.736	0.736	0.702	0.736	0.736	0.702	0.717	0.717	0.561
bu	0.579	0.527	0.536	0.579	0.527	0.536	0.563	0.520	0.512
c	0.762	0.867	0.684	0.762	0.867	0.684	0.773	0.870	0.698
d	0.935	0.935	0.938	0.935	0.935	0.938	0.934	0.934	0.936
e	0.825	0.827	0.825	0.414	0.423	0.414	0.110	0.167	0.106
h	0.637	0.691	0.657	0.637	0.691	0.657	0.480	0.581	0.491
io	0.862	0.868	0.458	0.862	0.868	0.458	0.845	0.853	0.578
ir	0.965	0.961	0.978	0.947	0.942	0.968	0.945	0.943	0.968
ma	1.000	1.000	1.000	1.000	1.000	1.000	1.000	1.000	1.000
me	0.582	0.681	0.615	0.582	0.681	0.615	0.583	0.688	0.607
ph	0.771	0.767	0.774	0.771	0.767	0.774	0.720	0.718	0.724
pi	0.699	0.685	0.704	0.699	0.685	0.704	0.661	0.653	0.670
po	0.879	0.872	0.897	0.879	0.872	0.897	0.468	0.466	0.473
r	0.726	0.723	0.728	0.726	0.723	0.728	0.733	0.730	0.735
sb	0.711	0.718	0.712	0.711	0.718	0.712	0.686	0.694	0.686
se	0.924	0.921	0.926	0.924	0.921	0.926	0.520	0.516	0.497
te	0.889	0.890	0.892	0.389	0.392	0.408	0.393	0.387	0.404
th	0.983	0.981	0.982	0.974	0.972	0.973	0.849	0.825	0.851
ti	0.788	0.778	0.681	0.788	0.778	0.681	0.752	0.732	0.405
tw	0.717	0.714	0.724	0.717	0.714	0.724	0.717	0.714	0.724
wd	0.902	0.893	0.918	0.902	0.893	0.918	0.893	0.884	0.911
wi	0.936	0.955	0.944	0.936	0.955	0.944	0.931	0.951	0.941
wr	0.831	0.827	0.823	0.493	0.481	0.468	0.241	0.227	0.225
ww	0.839	0.838	0.840	0.459	0.457	0.464	0.205	0.224	0.208
y	0.866	0.861	0.865	0.349	0.325	0.344	0.223	0.214	0.234
rank	2.00	2.14	1.61	2.00	2.14	1.61	1.93	2.04	1.79

**Table 3 entropy-22-01129-t003:** Micro-average precision and recall for the random forest, the majority voting and the proposed algorithm together with Friedman ranks.

	Precision_*μ*_			Recall_*μ*_		
Dataset	$\Psi_{\mathrm{mv}}$	$\Psi_{\mathrm{rf}}$	$\Psi_{i}$	$\Psi_{\mathrm{mv}}$	$\Psi_{\mathrm{rf}}$	$\Psi_{i}$
aa	0.469	0.475	0.477	0.469	0.473	0.477
ap	0.853	0.812	0.863	0.853	0.812	0.863
ba	0.683	0.712	0.722	0.683	0.712	0.722
bi	0.736	0.736	0.702	0.736	0.736	0.702
bu	0.579	0.527	0.536	0.579	0.527	0.536
c	0.762	0.867	0.684	0.762	0.867	0.684
d	0.935	0.935	0.938	0.935	0.935	0.938
e	0.418	0.424	0.417	0.411	0.423	0.411
h	0.637	0.691	0.657	0.637	0.691	0.657
io	0.862	0.868	0.458	0.862	0.868	0.458
ir	0.947	0.942	0.968	0.947	0.942	0.968
ma	1.000	1.000	1.000	1.000	1.000	1.000
me	0.582	0.681	0.615	0.582	0.681	0.615
ph	0.771	0.767	0.774	0.771	0.767	0.774
pi	0.699	0.685	0.704	0.699	0.685	0.704
po	0.879	0.872	0.897	0.879	0.872	0.897
r	0.726	0.723	0.728	0.726	0.723	0.728
sb	0.711	0.718	0.712	0.711	0.718	0.712
se	0.924	0.921	0.926	0.924	0.921	0.926
te	0.389	0.392	0.408	0.389	0.392	0.408
th	0.974	0.972	0.973	0.974	0.972	0.973
ti	0.788	0.778	0.681	0.788	0.778	0.681
tw	0.717	0.714	0.724	0.717	0.714	0.724
wd	0.902	0.893	0.918	0.902	0.893	0.918
wi	0.936	0.955	0.944	0.936	0.955	0.944
wr	0.493	0.481	0.468	0.493	0.481	0.468
ww	0.459	0.457	0.464	0.459	0.457	0.464
y	0.350	0.325	0.344	0.349	0.325	0.344
rank	2.00	2.14	1.64	2.00	2.14	1.61

**Table 4 entropy-22-01129-t004:** Macro-average precision and recall for the random forest, the majority voting and the proposed algorithm together with Friedman ranks.

	Precision_*μ*_			Recall_*μ*_		
Dataset	$\Psi_{\mathrm{mv}}$	$\Psi_{\mathrm{rf}}$	$\Psi_{i}$	$\Psi_{\mathrm{mv}}$	$\Psi_{\mathrm{rf}}$	$\Psi_{i}$
aa	0.179	0.171	0.152	0.217	0.218	0.209
ap	0.705	0.557	0.710	0.663	0.563	0.684
ba	0.475	0.475	0.486	0.491	0.512	0.519
bi	0.712	0.712	0.526	0.721	0.721	0.614
bu	0.564	0.521	0.513	0.561	0.520	0.511
c	0.779	0.872	0.702	0.767	0.868	0.693
d	0.936	0.935	0.938	0.932	0.933	0.935
e	0.079	0.123	0.075	0.182	0.259	0.186
h	0.474	0.594	0.485	0.486	0.568	0.500
io	0.860	0.881	0.596	0.831	0.828	0.562
ir	0.944	0.942	0.968	0.945	0.944	0.968
ma	1.000	1.000	1.000	1.000	1.000	1.000
me	0.582	0.683	0.614	0.585	0.692	0.600
ph	0.726	0.721	0.730	0.715	0.716	0.718
pi	0.664	0.652	0.670	0.659	0.655	0.669
po	0.451	0.450	0.451	0.487	0.483	0.496
r	0.742	0.738	0.743	0.724	0.721	0.726
sb	0.723	0.728	0.721	0.653	0.663	0.655
se	0.542	0.527	0.495	0.504	0.507	0.501
te	0.397	0.380	0.400	0.390	0.393	0.409
th	0.816	0.803	0.826	0.886	0.848	0.879
ti	0.853	0.780	0.341	0.673	0.692	0.500
tw	0.718	0.714	0.724	0.717	0.714	0.724
wd	0.894	0.883	0.911	0.892	0.886	0.911
wi	0.926	0.948	0.931	0.935	0.954	0.951
wr	0.246	0.228	0.234	0.236	0.226	0.216
ww	0.226	0.247	0.230	0.189	0.206	0.191
y	0.232	0.200	0.244	0.216	0.230	0.226
rank	1.86	2.07	1.79	2.18	1.82	1.82

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
