# Peer review of "Decision Tree Integration Using Dynamic Regions of Competence"

_entropy, 2020, doi:10.3390/e22101129_

Round 1

Reviewer 1 Report

Intelligent decision-making is important. The topic of the manuscript is interesting. Some comments are helpful for improving the work. 1. Experimental setup can be improved. The logic of the current 2 paragraphs in the Experimental setup can be improved. 2. Both ‘5. Results’ and ‘6. Discussion’ can be improved. Current work is more like a short communication. As a full-length research article, the experiment, result, and discussion should be described more clearly and fruitfully. 3. The following work can be discussed in Conclusions.

Author Response

We are greatful for reviewing our article and hope, that our changes improve the clarity and readablity of our work.

1. The Experimental setup secion was split into more paragraphs and edited for greater quality. Additional comments were incorporated for the most difficult parts.
2. Additional experimental results were added to enhance the quality of the work and make the discussion more meaningful.
3. The discussion was extended in sections Results and Conclusions.

Once again we are thankful for pointing out the weakest parts of the article. We believe, that the changes we have added allow for better understanding of our work.

Reviewer 2 Report

Thanks for recommending me as a reviewer. This study proposed the algorithm that uses partitioning the feature space which split is determined by the decision rules of each decision tree node which is the base classification model. After dividing the feature space, the centroid of each new subspace is determined. The study was well written throughout. If the authors complete minor revisions, the quality of this study will improve.

  1. (minor revision). Author information should be more specific. Please, add the corresponding author's information.

2. Line 24-26: If the author provides more theoretical background to the "weighted majority voting rule" in the introductory section, it may help readers understand.

3. The experimental procedure was well described in detail.

4. Line 155-160: It is necessary to add a reason why the author used "F-score" as a way to evaluate the model's performance. 

5. It is well known that the ensemble model is highly accurate. Nevertheless, this study is meaningful as an empirical study.

Author Response

1. We have mistakenly omitted the correspondence details. It was corrected.
2. The introduction section has been rewritten to describe weighing issues.
3. We appreciate this commentary.
4. The reason why F-score was used was described in the previous section Experimental setup and only indicated in the Results section. For the greater clarity we have stressed the link to the quality measure in the latter section.
5. We are thankful for noticing that. We believe that decision tree ensembling is a perfect field of study and new techniques and algorithms can lead to better understanding and quality of classification process.

Reviewer 3 Report

The author must include an appendix on tree topologies, at least the more relevant used in the method construction for machine intelligence and artificial intelligence. The appendix could be short. In the introduction must be included another existing methods and why these methods are non-efficient, or no effectives. Also, if is possible include a comparative analysis of this efficiency and efficacy. The authors must improve the conclusions, given the research prospectives of the method. The authors must include also in other appendix the theorem or lemmas that give support of the method and its imporovement. The author included correctly an entropy distribution or stochastic model of certainty of the methods. This is interesting.

Author Response

We are greatful for reading out article and pointing out the possible improvements.

1. The Experimental setup was extended with a description of decision tree implementation details. Additionaly to provide the most accurate information about the tree topology, the code was published under https://github.com/TAndronicus/dynamic-dtree.
2. The introductory section has been rewritten.
3. Also, a comparative analysis this efficiency and efficacy was included. This goal was achieved using the statistical tests described in the paper.
4. The Conclusions section has been rewritten.
5. Appendix with theorem or lemmas.
Probably the theorem or lemmas would strengthen the proposed method. This is a true and correct remark about our approach. However, for the time being, we cannot formulate an appropriate theorem or lemma.

We hope, that the changes we have incorporated will improve the quality of the article and provide better understanding of the topic and the conducted experiments.

Round 2

Reviewer 3 Report

The authors could improve the redaction in some parts of the paper.